# Pregnancy success in mice requires appropriate cannabinoid receptor signaling for primary decidua formation

Yingju Li[1,2†], Amanda Dewar[1,2†], Yeon Sun Kim[1,2], Sudhansu K Dey[1,2*], Xiaofei Sun[1,2*]

[1]Division of Reproductive Sciences, Cincinnati Children's Hospital Medical Center, Cincinnati, United States; [2]Department of Pediatrics, University of Cincinnati College of Medicine, Cincinnati, United States

**Abstract** With implantation, mouse stromal cells begin to transform into epithelial-like cells surrounding the implantation chamber forming an avascular zone called the primary decidual zone (PDZ). In the mouse, the PDZ forms a transient, size-dependent permeable barrier to protect the embryo from maternal circulating harmful agents. The process of decidualization is critical for pregnancy maintenance in mice and humans. Mice deficient in cannabinoid receptors, CB1 and CB2, show compromised PDZ with dysregulated angiogenic factors, resulting in the retention of blood vessels and macrophages. This phenotype is replicated in *Cnr1*[-/-] but not in *Cnr2*[-/-] mice. In vitro decidualization models suggest that *Cnr1* levels substantially increase in mouse and human decidualizing stromal cells, and that neutralization of CB1 signaling suppresses decidualization and misregulates angiogenic factors. Taken together, we propose that implantation quality depends on appropriate angiogenic events driven by the integration of CB2 in endothelial cells and CB1 in decidual cells.

**\*For correspondence:**
sk.dey@cchmc.org (SKD);
xiaofei.sun@cchmc.org (XS)

[†]These authors contributed equally to this work

**Competing interests:** The authors declare that no competing interests exist.

## Introduction

Decidualization is a critical pregnancy event that follows embryo implantation. During the initial stages of decidualization in mice, stromal cells transform into epithelioid cells (epithelial-like) surrounding the implanting embryo and form an avascular zone. This region is called the primary decidual zone (PDZ) and is critical to embryo development and successful pregnancy outcomes in mice (*Cha et al., 2012*). The PDZ encircles the implantation chamber (crypt) on day 5 of pregnancy and is fully established by day 6. Because the PDZ is avascular, this zone is normally devoid of maternal immune cells, and protects embryos from circulating maternal insults (*Tung et al., 1986*). Recently we have shown that Scribble (Scrib), a scaffold protein and a component of the planar cell polarity (PCP), plays a key role in PDZ formation (*Yuan et al., 2019*). However, molecular regulation in the formation of the PDZ is still far from clear.

Natural cannabinoids (tetrahydrocannabinol, the most potent psychoactive component) and endocannabinoids (anandamide and 2-arachidonoilglycerol) signal through two membrane receptors, CB1 (encoded by *Cnr1*) and CB2 (encoded by *Cnr2*). Both CB1 and CB2 are G-protein coupled receptors in the Gi/o and Gq families (*Pertwee et al., 2010*). CB1 is mainly expressed in the central nervous system (CNS), but also appears in peripheral tissues including the uterus, heart, testis, liver, and small intestine. By contrast, CB2 is mostly expressed in immune cells (*Berdyshev, 2000*). We have recently shown that CB2 is also expressed in uterine endothelial cells (*Li et al., 2019*). Deletion of *Cnr2* increases uterine edema before implantation and causes suboptimal implantation (*Li et al., 2019*).

Our work has previously shown that elevated cannabinoid signaling is detrimental to early pregnancy events in mice (*Sun and Dey, 2012*). Furthermore, we have shown that *Cnr1^{-/-}Cnr2^{-/-}* mice have compromised implantation (*Li et al., 2019*), suggesting that silencing of endocannabinoid signaling is also detrimental to early pregnancy events. In the study, we showed that the litter sizes of double mutant mice, but not single mutant mice, are substantially smaller than wild-type (WT) mice. We also found that endothelial cells with *Cnr2* deletion play an important role in embryo implantation (*Li et al., 2019*). Interestingly, resorption rate is lower in *Cnr2^{-/-}* females than in *Cnr1^{-/-}Cnr2^{-/-}* females during the midgestational stage, indicating a critical role of CB1 in early pregnancy. Since CB1 abundance is quite low in the uterus, its role in early pregnancy remains obscure.

In this study, we investigated the roles of endocannabinoid signaling in decidualization using mouse models with suppressed CB1 and CB2 and a primary culture of *Cnr1^{-/-}* stromal cells. The results show that double mutant mice have compromised PDZ formation, resulting in suboptimal pregnancy outcomes. Further investigation revealed that *Cnr1* is expressed in decidual cells. These results, combined with our previous knowledge of CB2 expression in endothelial cells, show for the first time that angiogenic actions incited by CB2 are integrated by CB1 in decidual cells to form the avascular PDZ, which is critical for pregnancy success.

## Results

### *Cnr1^{-/-}Cnr2^{-/-}* females have suboptimal decidualization

Previously we showed that *Cnr1^{-/-}Cnr2^{-/-}* uteri have defective decidualization and consequent increases in midgestational resorption rates. However, the mechanism by which the absence of CB1 and CB2 signaling compromises uterine decidualization remains unclear. To explore this further, WT and *Cnr1^{-/-}Cnr2^{-/-}* females were mated with WT males. Induction of *Ptgs2*, a rate-limiting enzyme for prostaglandin formation, at the crypt (implantation chamber) epithelium and underlying stroma has been shown to be critical to initiate decidualization (*Lim et al., 1997*). Thus, we examined the expression of *Ptgs2* on day 5 of pregnancy to capture the influence of *Ptgs2*. The expression pattern and signal intensity of *Ptgs2* is comparable in WT and *Cnr1^{-/-}Cnr2^{-/-}* females (*Figure 1a*), indicating stimulation from embryo implantation is well received by *Cnr1^{-/-}Cnr2^{-/-}* uteri. However, the in situ hybridization of *Bmp2*, a morphogen expressed by cells undergoing decidualization at the site of embryo implantation (*Lee et al., 2007*; *Paria et al., 2001*), reveals that stromal cell decidualization is defective in *Cnr1^{-/-}Cnr2^{-/-}* females on day 5 of pregnancy (*Figure 1b*). Expression of *Hoxa10*, also critical for decidualization (*Benson et al., 1996*), shows a similar reduction in *Cnr1^{-/-}Cnr2^{-/-}* implantation sites (*Figure 1—figure supplement 1*). Decidual defects in *Cnr1^{-/-}Cnr2^{-/-}* uteri continue on day 6 of pregnancy. The tridimensional (3D) structure of implantation sites reveals that, on day 6 of pregnancy, decidual responses in WT mice promote the uterine lumen to form an arch shape, whereas in *Cnr1^{-/-}Cnr2^{-/-}* implantation sites, the lumen remains flat (*Figure 2a*). The domain of *Bmp2* positive signals is reduced in *Cnr1^{-/-}Cnr2^{-/-}* implantation sites (*Figure 2b*). In contrast, stromal cells after decidual transformation downregulate the expression of *Bmp2*. More decidualized cells around the implantation chamber negative for *Bmp2* signals are observed on day 6 of pregnancy in WT mice as compared to *Cnr1^{-/-}Cnr2^{-/-}* implantation sites (*Figure 2b*). With decidualization in progress, *Ptgs2* expression transitions from the antimesometrial side of the crypt to the mesometrial side in the WT crypt on day 6 of pregnancy. In *Cnr1^{-/-}Cnr2^{-/-}* implantation sites, cells on the lateral side of the implantation chamber show *Ptgs2* signals, suggesting impeded progress of decidualization (*Figure 2c*).

### Macrophages are retained around the implantation chambers of compromised deciduae in *Cnr1^{-/-}Cnr2^{-/-}* females

Hematopoietic cells, including macrophages, are distributed throughout mouse uteri before implantation, though these cells are not present in the decidual zone after implantation (*Erlebacher, 2013*; *Nancy et al., 2012*). We observed that the distribution of hematopoietic cells, as revealed by a pan hematopoietic marker CD45, is comparable in *Cnr1^{-/-}Cnr2^{-/-}* and WT uteri on day 4 of pregnancy (*Figure 3a*). This observation is further confirmed by the distribution of macrophages, which are demarcated by a macrophage marker F4/80 staining (*Austyn and Gordon, 1981*; *Figure 3b*). These results suggest that hematopoietic cells normally enter the uterus of *Cnr1^{-/-}Cnr2^{-/-}* mice before

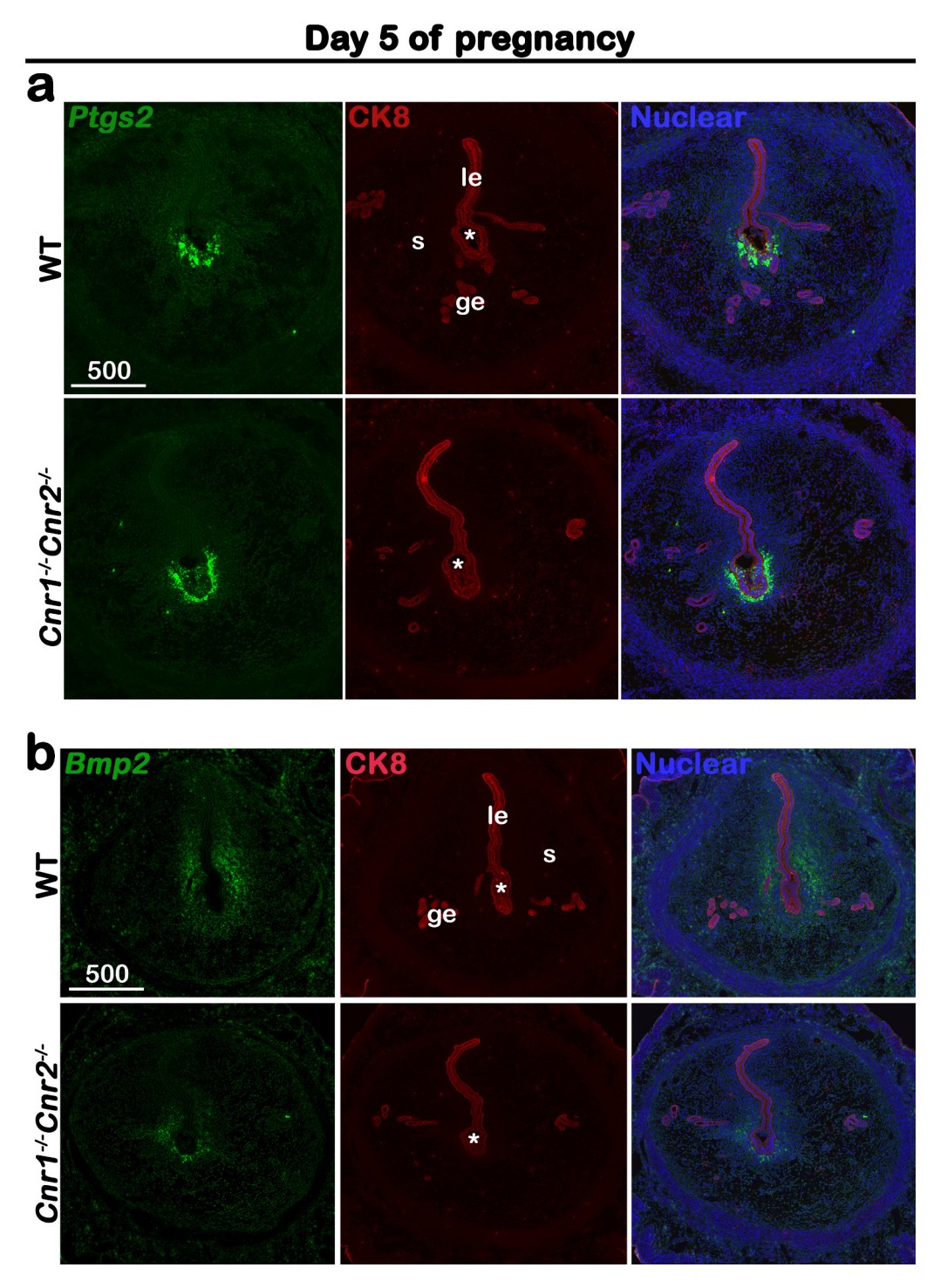

**Figure 1.** $Cnr1^{-/-}Cnr2^{-/-}$ females show normal responses to embryonic stimulation during implantation, but decidual process is compromised. (**a**) In situ hybridization of $Ptgs2$ in uteri on day 5 of pregnancy. The expression patterns of $Ptgs2$ critical for implantation show no significant difference in WT and $Cnr1^{-/-}Cnr2^{-/-}$ females. CK8 staining outlines uterine epithelial cells. (**b**) In situ hybridization of $Bmp2$ in uteri on day 5 of pregnancy. Decidual responses in $Cnr1^{-/-}Cnr2^{-/-}$ females are much weaker than those in WT females. le, luminal epithelium; s, stroma; ge, glandular epithelium; Asterisks, positions of embryos; Scale bars, 500 µm. All images are representative of three independent experiments.

The online version of this article includes the following figure supplement(s) for figure 1:

**Figure supplement 1.** In situ hybridization of $Hoxa10$ in uteri on day 5 of pregnancy.

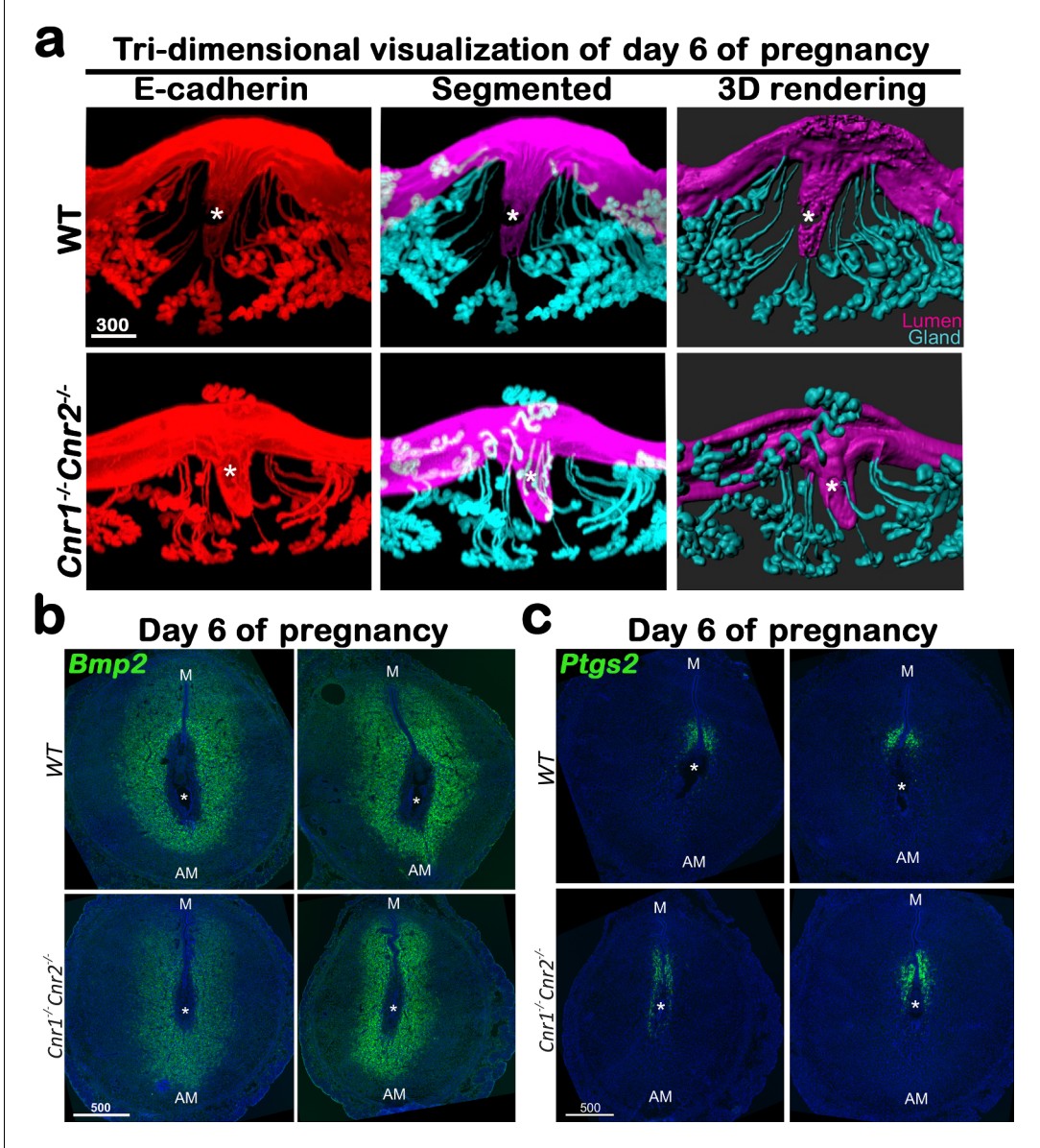

**Figure 2.** *Cnr1⁻ᐟ⁻Cnr2⁻ᐟ⁻* females have suboptimal decidualization. (a) 3D visualization of day 6 implantation sites in WT and *Cnr1⁻ᐟ⁻Cnr2⁻ᐟ⁻* females. Images of E-cadherin immunostaining, segmented, and 3D rendered images of day 6 implantation sites in each genotype show compromised decidual responses in *Cnr1⁻ᐟ⁻Cnr2⁻ᐟ⁻* females. Scale bars, 300 μm. (b and c) In situ hybridization of *Bmp2* and *Ptgs2* in uteri on day 6 of pregnancy. M, mesometrial; AM, anti-mesometrial; Asterisks, positions of embryos; Scale bars, 500 μm. All images are representative of three independent experiments.

implantation. On day 6 of pregnancy, most hematopoietic cells migrate away from the implantation chamber in WT females, whereas some hematopoietic cells fail to do so in *Cnr1⁻ᐟ⁻Cnr2⁻ᐟ⁻* females (*Figure 3—figure supplement 1*). Similarly, most macrophages are absent from the WT implantation chamber, but some macrophages remain around the *Cnr1⁻ᐟ⁻Cnr2⁻ᐟ⁻* embryos, albeit at reduced number (*Figure 3c*). The significant lingering of macrophages in the *Cnr1⁻ᐟ⁻Cnr2⁻ᐟ⁻* endometrium (*Figure 3d*) suggests that impeded decidual responses in *Cnr1⁻ᐟ⁻Cnr2⁻ᐟ⁻* females fail to dispel macrophages efficiently. Collectively, the result suggests that the normally developed decidua protects the developing blastocyst from inflammatory insult from macrophages and that this safe-guard mechanism is compromised in double-mutant mice.

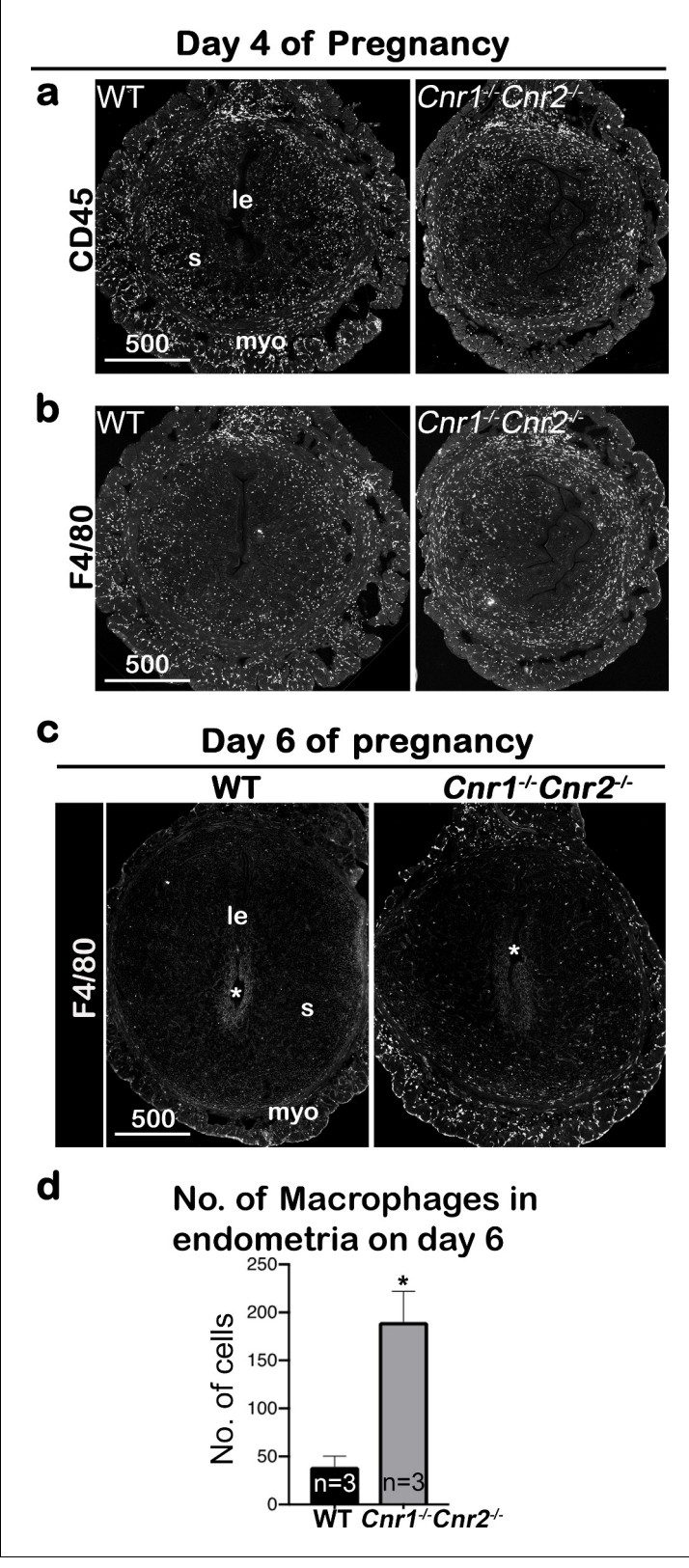

**Figure 3.** Macrophages are retained around implantation chambers of compromised deciduae in *Cnr1⁻/⁻Cnr2⁻/⁻* females. (a) Uterine leukocytes in pre-implantation are highlighted by CD45, which is expressed on all leukocytes. (b) Immunofluorescent staining of F4/80 on day 4 of pregnancy to mark macrophages. (c) Immunofluorescent staining of F4/80 on day 6 of pregnancy. (d) The numbers of macrophages within the endometrial domains are

*Figure 3 continued on next page*

*Figure 3 continued*

quantified using three sections obtained from three different animals in each genotype. le, luminal epithelium; s, stroma; myo, myometrium; Asterisks, positions of embryos; Scale bars, 500 μm; *p<0.05, *Student's t*-tests. All images are representative of three independent experiments.

The online version of this article includes the following figure supplement(s) for figure 3:

**Figure supplement 1.** Immunofluorescence of CD45 on day 6 of pregnancy.

## The formation of the primary decidual zone is compromised in *Cnr1$^{-/-}$Cnr2$^{-/-}$* implantation sites

On day 6 of pregnancy, the first stratum of the decidual zone, called the primary decidual zone (PDZ), is fully established. As previously stated, the PDZ is an avascular zone comprised of stromal cells undergoing epithelial-like transformation and is impermeable to maternal immune cells; as such, this zone is indispensable to protect and support further embryo development (*Tung et al., 1986*). The finding that macrophages remain close to *Cnr1$^{-/-}$Cnr2$^{-/-}$* implantation chambers prompted us to examine the formation of the PDZ on day 6 of pregnancy. PDZ formation requires Scrib signaling (*Yuan et al., 2019*), yet the Scrib positive domain is much reduced in *Cnr1$^{-/-}$Cnr2$^{-/-}$* females (*Figure 4a*). This result is also supported by the presence of blood vessels, revealed by immunostaining of FLK1, a reliable marker of blood vessels, in the *Cnr1$^{-/-}$Cnr2$^{-/-}$* PDZ on day 6 of pregnancy (*Figure 4b*). The presence of blood vessels in *Cnr1$^{-/-}$Cnr2$^{-/-}$* PDZ is significantly abundant than that in WT PDZ (*Figure 4—figure supplement 1*). *Vegfa*, which promotes angiogenesis by targeting its receptor FLK1 (*Millauer et al., 1993*), shows a dynamic spatiotemporal expression pattern in periimplantation uteri (*Halder et al., 2000*). *Vegfa* expression is downregulated in WT PDZ on day 6 of pregnancy, whereas positive *Vegfa* signals are observed in the PDZ of *Cnr1$^{-/-}$Cnr2$^{-/-}$* mice (*Figure 4b*), similar to FLK1 positive cells in the same area. These results indicate that normal disappearance of blood vessels is hindered in *Cnr1$^{-/-}$Cnr2$^{-/-}$* PDZ, accompanied by lingering macrophages around the embryo.

## Retention of blood vessels in *Cnr1$^{-/-}$Cnr2$^{-/-}$* PDZ is associated with multiple angiogenic factors

Proangiogenic angiopoietin-1 (ANGPT1), targeting TIE2 receptors on endothelia, is critical to assemble newly formed vasculature and stabilize the vascular network and integrity (*Davis et al., 1996*; *Suri et al., 1996*). By contrast, angiopoietin-2 (ANGPT2), an ANGPT1 antagonist (*Maisonpierre et al., 1997*), promotes endothelial cell apoptosis and vascular regression. *Angpt1* is expressed in most stromal/decidual cells but is absent in the PDZ region of day 6 implantation sites in WT mice (*Figure 5a*). However, *Angpt1* expression remains in the PDZ of *Cnr1$^{-/-}$Cnr2$^{-/-}$* mice, which facilitates the stability of vessels in the region (*Figure 5a*). The signal intensity of *Angpt1* is quantified in *Figure 5—figure supplement 1*. In contrast to *Angpt1*, *Angpt2* is expressed in decidualized stromal cells surrounding the implantation chamber at the mesometrial side in WT females (*Figure 5b*). The domain expressing *Angpt2* and the signal intensity are much reduced in *Cnr1$^{-/-}$Cnr2$^{-/-}$* implantation sites (*Figure 5b* and *Figure 5—figure supplement 1*). These results indicate that opposing functions between ANGPT1 and ANGPT2 at implantation sites maintain a vascular network that supports normal implantation; in *Cnr1$^{-/-}$Cnr2$^{-/-}$* implantation sites, these opposing functions are dysregulated. To further determine the roles of CB1 and CB2 in the regulation of ANGPTs, RNA levels of *Angpts* were examined in *Cnr1$^{-/-}$* and *Cnr2$^{-/-}$* mice. Intriguingly, the expression pattern of *Angpt1* and *Angpt2* in *Cnr1$^{-/-}$* implantation sites recapitulate their patterns in *Cnr1$^{-/-}$Cnr2$^{-/-}$* counterparts (*Figure 5a and b*), suggesting that defects in ANGPT regulation are due to CB1 deficiency.

Hypoxia-inducible factors (HIFs) are transcriptional regulators of angiopoietins and vascular endothelial growth factors (VEGFs; *Ortiz-Barahona et al., 2010*; *Benita et al., 2009*). Interestingly, we found the expression pattern of *Hif1a* is similar to that of *Angpt1*, showing broad expression in stromal cells but lacking expression in WT PDZs on day 6 of pregnancy (*Figure 5c*). On the other hand, the pattern of *Hif2a* expression is similar to that of *Angpt2* with positive signals in stromal cells close to implantation sites (*Figure 5d*). PDZs in *Cnr1$^{-/-}$* and *Cnr1$^{-/-}$Cnr2$^{-/-}$* mice show increased *Hif1a* expression, as well as decreased *Hif2a* expression compared with WT females on day 6 of pregnancy

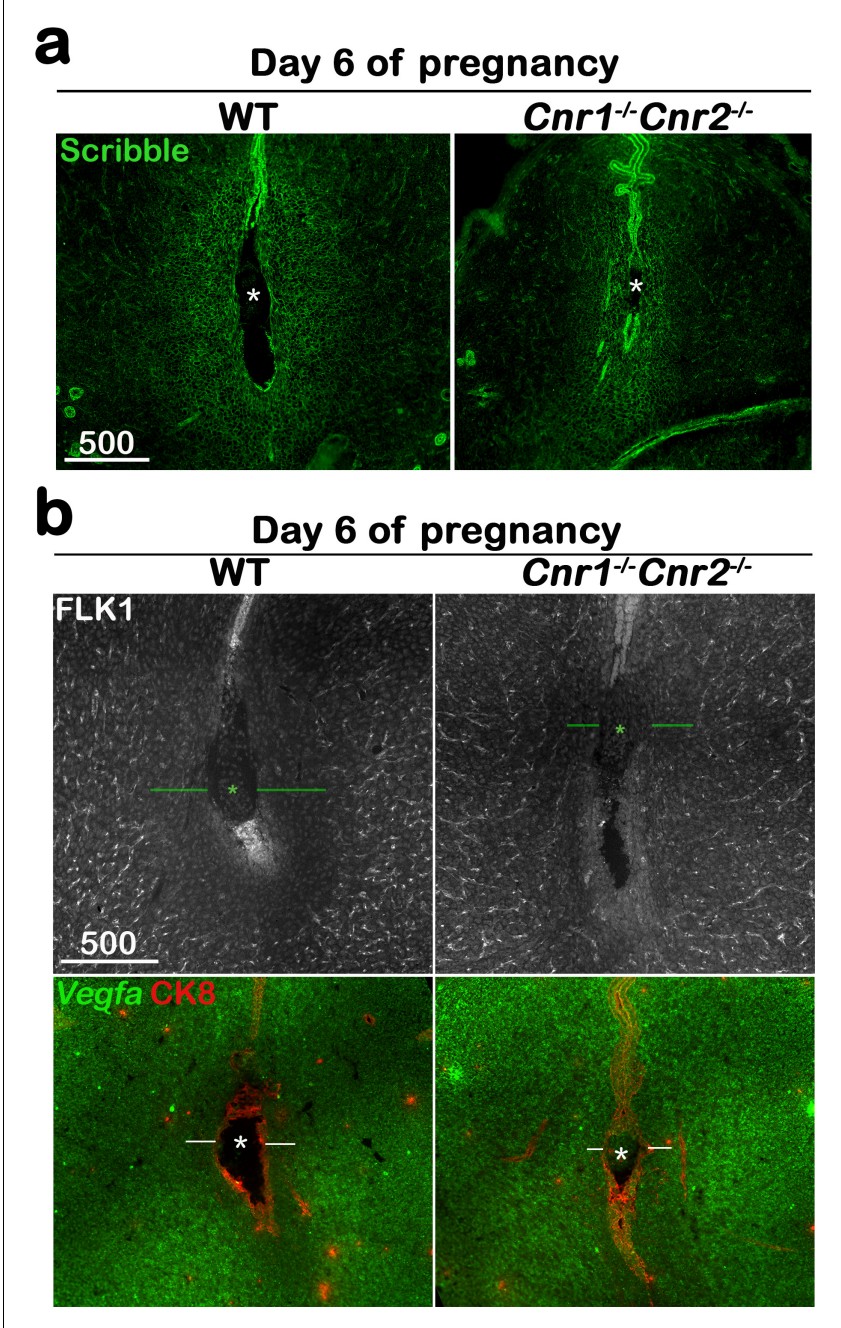

**Figure 4.** The formation of the primary decidual zone is compromised in *Cnr1⁻/⁻Cnr2⁻/⁻* implantation sites. (a) The primary decidual zone is highlighted by Scribble on day 6 of pregnancy. (b) Immunofluorescence staining of FLK1 and in situ hybridization of *Vegfa* in day 6 pregnant uteri. Blood vessels and higher *Vegfa* signals are observed in the PDZ area (indicated by green or white lines) of *Cnr1⁻/⁻Cnr2⁻/⁻* implantation sites. CK8 staining outlines epithelial cells. Asterisks, positions of embryos; Scale bars, 500 μm. All images are representative of three independent experiments.

The online version of this article includes the following figure supplement(s) for figure 4:

**Figure supplement 1.** Quantitation of blood vessel area in PDZs on day 6 of pregnancy.

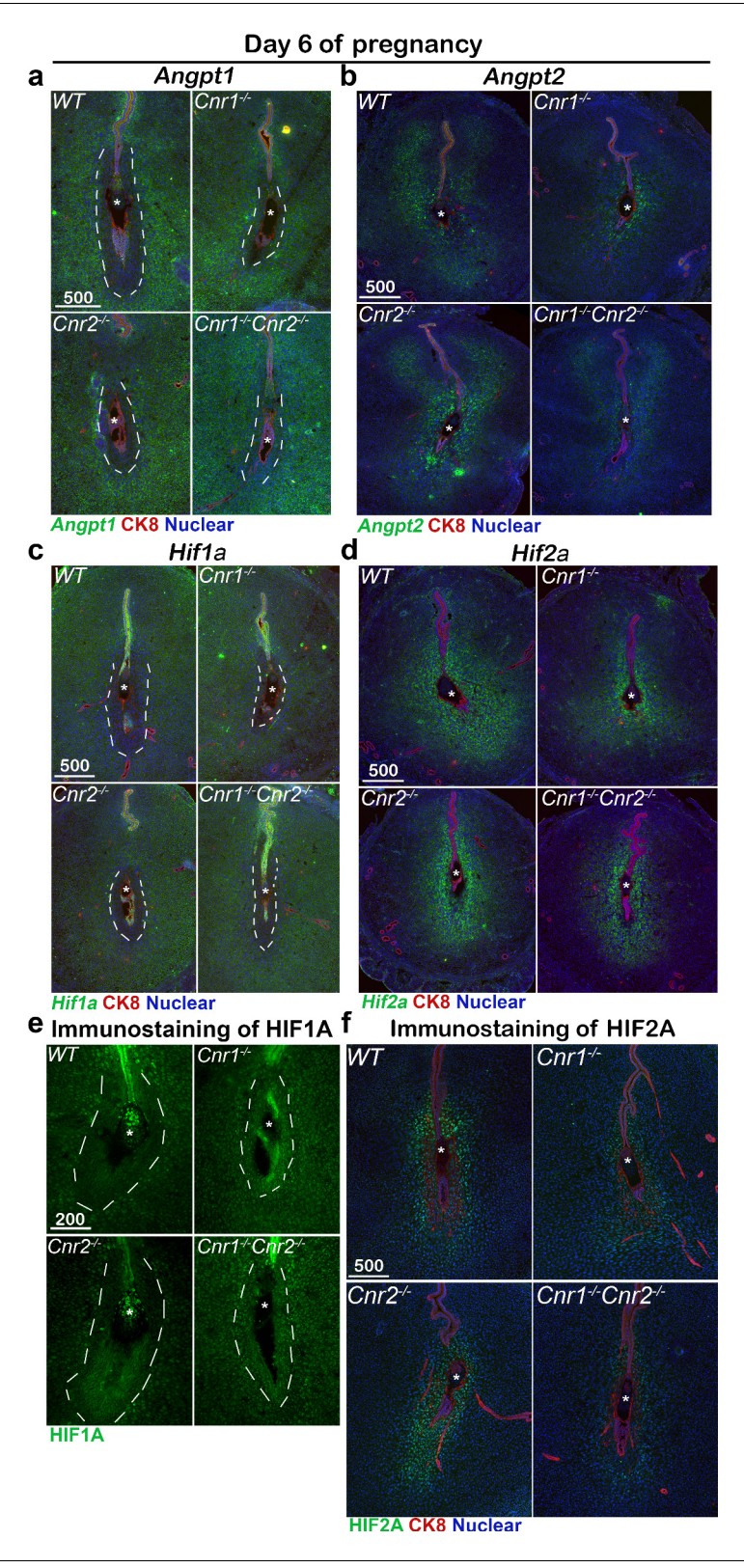

**Figure 5.** Angiogenic factors are misregulated in *Cnr1*<sup>-/-</sup>*Cnr2*<sup>-/-</sup> and *Cnr1*<sup>-/-</sup> PDZ. (a–d) In situ hybridization of *Angpt1*, *Angpt2*, *Hif1a* and *Hif2a* on day 6 of pregnancy. CK8 staining outlines uterine epithelial cells. (e and f) Immunostaining of HIF1A and HIF2A on day 6 of pregnancy. Dotted lines outline the PDZs. Asterisks, positions of embryos; Scale bars, 500 μm. All images are representative of three independent experiments.

*Figure 5 continued on next page*

*Figure 5 continued*

The online version of this article includes the following figure supplement(s) for figure 5:

**Figure supplement 1.** Quantification of signals of angiogenic factors on day 6 of pregnancy.

(*Figure 5c and d*, *Figure 5—figure supplement 1*). The mRNA expression patterns of *Hif1a* and *Hif2a* were further confirmed by immunostaining using antibodies specific to *Hif1a* and *Hif2a* (*Figure 5e and f*). These results suggest that increased *Angpt1* expression with decreased *Angpt2* expression in the PDZs of *Cnr1*[-/-]*Cnr2*[-/-] stabilizes blood vessels in this region. The differential expression patterns of *Hif1a* and *Hif2a* indicate that *Angpt1* and *Angpt2* are perhaps differentially regulated by Hif1a and HIF2A, respectively. The similar patterns of angiogenic factors in *Cnr1*[-/-] and *Cnr1*[-/-]*Cnr2*[-/-] mice suggest that CB1 plays a key role in PDZ formation.

## *Cnr1* is induced in mouse and human decidual cells

CB2 is primarily expressed in immune cells. Using a *Cnr2* reporter mouse line (*López et al., 2018*), we have recently shown that uterine endothelial cells express CB2 (*Li et al., 2019*). Studies show that the deletion of *Cnr1* in the mouse uterus compromises pregnancy outcomes (*Wang et al., 2006*), even though *Cnr1* levels are quite low in non-pregnant uterine cells. We observed that *Cnr1* levels increase in implantation sites from day 4 to day 8 (the peak of decidual response; *Figure 6a*). Since most uterine epithelial cells are removed in implantation sites on day 8, we attribute the increase to changes in stromal/decidual cells. We isolated and purified uterine stromal cells on day 4 of pregnancy and induced them to decidualize in vitro by hormone treatments according to our established protocol (*Sun et al., 2016*). Levels of *Cnr1* increased about threefold in WT and *Cnr2*[-/-] decidualizing cells as compared to non-decidualized stromal cells in each genotype (*Figure 6b*). We did not include levels of Cnr1 from *Cnr1*[-/-] mice because *Cnr1* levels are undetectable in *Cnr1*[-/-] mice.

To study the roles of *Cnr1* during decidualization, stromal cells collected from both *Cnr1*[-/-] and WT uteri were induced to a decidual reaction *in vitro*. *Prl8a2*, a marker gene expressed in mouse decidual cells, was significantly lower in *Cnr1*[-/-], but not *Cnr2*[-/-], decidual cells (*Figure 6c*), suggesting deletion of *Cnr1* in stromal cells compromises decidualization. Interestingly, the induction of *CNR1* is even more robust in human uterine fibroblast (Huf) cells when decidualized in vitro. A steady increase of *CNR1* was observed during the cell transformation, and eventually *CNR1* increased by more than hundredfold compared with cells before decidualization (*Figure 7a*). The decidual

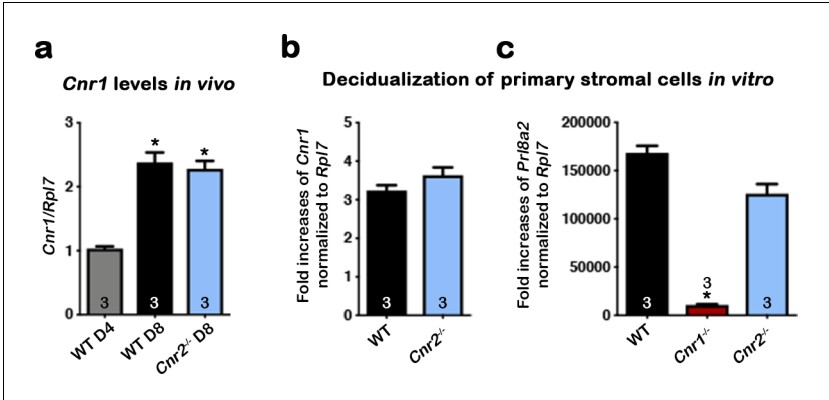

**Figure 6.** CB1 expression in mouse uterine stromal cells increases during decidualization. (a) Levels of *Cnr1* are higher at the peak of decidual response on day 8 of pregnancy as compared to day 4 of pregnancy. Samples were from three different females in each genotyping group. (b) *Cnr1* is induced in primary culture of WT and *Cnr2*[-/-] stromal cells undergoing decidualization in vitro. (c) Decidualization is severely reduced in primary *Cnr1*[-/-] stromal cells after decidual stimulation in vitro. Prl8a2 expression served as an indicator of decidual responses. Primary cells were collected from three different females in each genotyping groups. Values are mean ± SEM; Numbers on bars represent sample sizes; *p<0.05, Unpaired *t*-tests with Welch's correction.

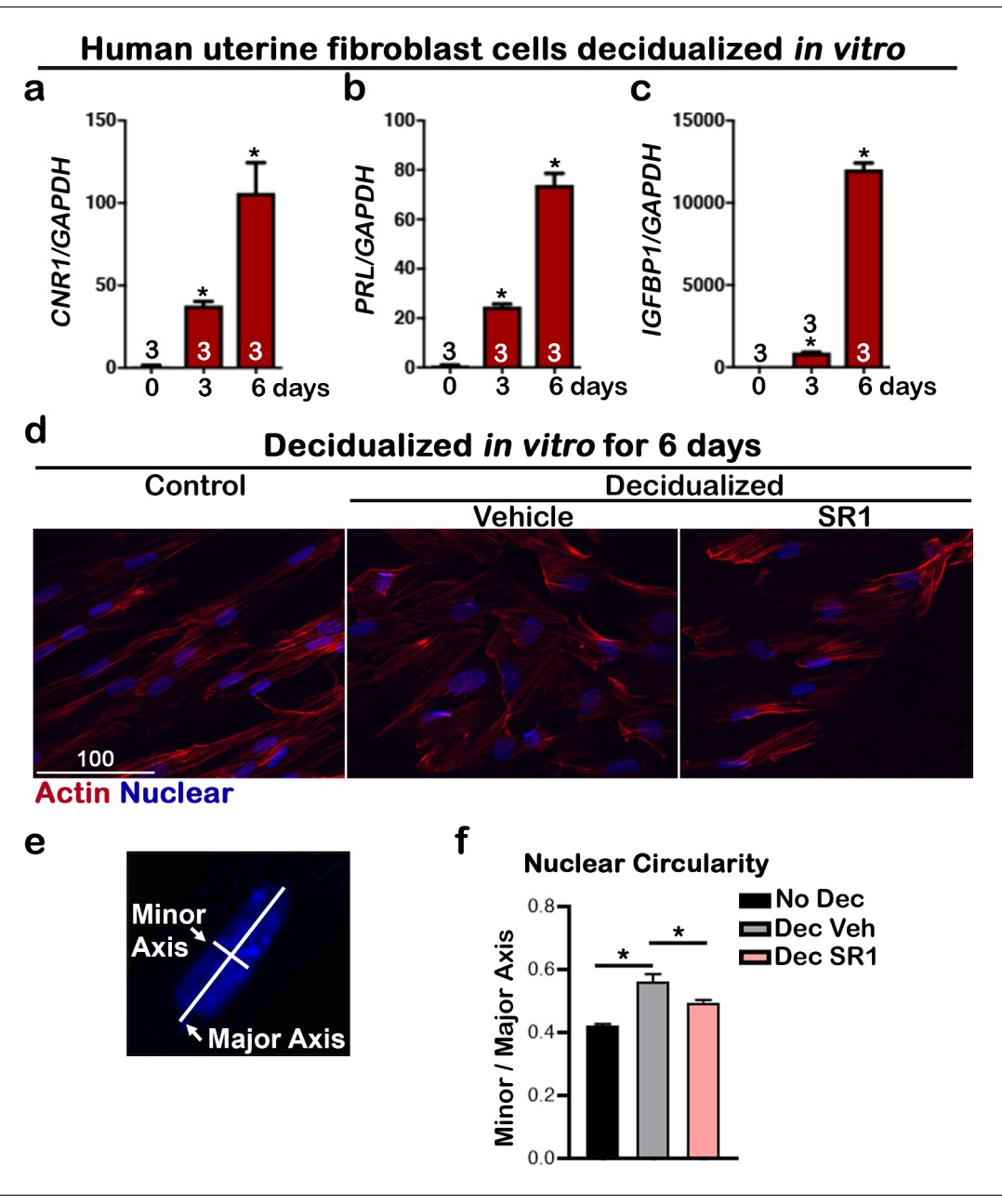

**Figure 7.** *CNR1* levels are upregulated in human uterine fibroblasts during in vitro decidualization. (a) Levels of *CNR1* gradually increase in Huf cells undergoing decidualization. (b and c) Decidual responses are marked by significant increases in *PROLACTIN* and *IGFBP1* levels in Huf cells. Sample sizes on bars are the number of culture wells. (d) Morphology of Huf cells changes from spindle shapes to more round shape; SR1 treatment compromises the morphological change in Huf cells. Cells are outlined with actin staining. Scale bar, 100 μm. All images are representative of three culture wells. (e) A scheme depicts the measurement of minor and major axes for calculating nuclear circularity. (f) The nuclear circularity of Huf cells before and after decidualization. SR1 treatment compromises the decidual process. Three fields were randomly chosen in three different culture wells. Values are mean ± SEM; Numbers on bars are sample sizes; *p<0.05, Unpaired *t*-tests with Welch's correction.
The online version of this article includes the following figure supplement(s) for figure 7:

**Figure supplement 1.** Morphology of Huf cells before and after decidualization.
**Figure supplement 2.** *CNR2* shows no significant change in human uterine fibroblasts during in vitro decidualization.

responses in Huf cells in the 6 d after induction were confirmed by a rise in two decidual markers, *PROLACTIN* (*PRL*) and *IGFBP1* (*Figure 7b and c*). The morphology of Huf cells changed from a spindle shape to more of a round shape (*Figure 7d* and *Figure 7—figure supplement 1*), which was also reflected by increases in nuclear circularity during decidualization (*Figure 7e and f*). Cell circularity is calculated as a ratio of the minor axis versus the major axis (*Figure 7e*). The nuclear circularity in decidualized Huf cells is significantly higher than in cells before decidualization (*Figure 7f*). Although *CNR1* levels were hundredfold higher in decidualized cells compared to control cells (*Figure 7a*), *CNR2* levels show no significant changes (*Figure 7—figure supplement 2*), suggesting CB1 plays a more important role in human decidualization.

To further study the biological significance of the induction of *CNR1*, Huf cells were treated with a specific CB1 antagonist, SR141716 (SR1, 2 µM), during decidualization. The decidual process in SR1-treated Huf cells was significantly compromised, as revealed by changes in *PRL* and *IGFBP1* levels (*Figure 8a and b*), two established decidual markers. The levels of *VEGFA, ANGPT1, ANGPT2,* and *HIF2A* were significantly downregulated by SR1 treatment during decidualization (*Figure 8*), suggesting the induction of *CNR1* during decidualization is critical to this process. Taken together, the upregulation of *Cnr1* in decidual cells in both mice and humans plays a key role during decidual cell transformation as supported by both in vivo and in vitro studies.

## Discussion

In this study, we show that a complex interplay is orchestrated by cannabinoid signaling in decidualization by forming the PDZ along with changes in angiogenic features in the decidual bed. In $Cnr1^{-/-}Cnr2^{-/-}$ females, this organization is compromised due to defective PDZ formation and dysregulated angiogenic homeostasis. PDZ is avascular in WT implantation sites, whereas blood vessels are present in $Cnr1^{-/-}Cnr2^{-/-}$ PDZ, accompanied by an increased presence of macrophages.

Angiogenesis and vascular remodeling are hallmarks of implantation, decidualization, and placentation (*Halder et al., 2000*; *Matsumoto et al., 2018*; *Chakraborty et al., 1995*). Retention of blood

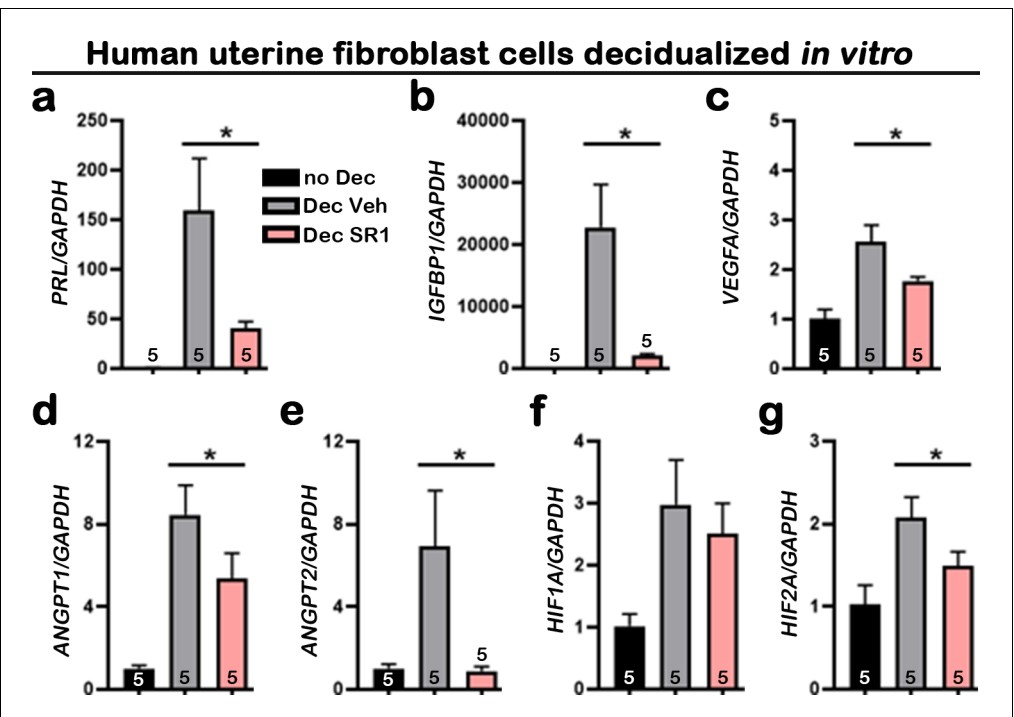

**Figure 8.** Suppression of CB1 impairs normal Huf cell decidualization and expression of angiogenic factors. (**a–g**) qPCR of *PRL, IGFBP1, VEGFA, ANGPT1, ANGPT2, HIF1A,* and *HIF2A* using RNA collected 3 d after decidualization. Values are mean ± SEM; Numbers on bars are numbers of culture wells; *p<0.05, Unpaired *t*-tests with Welch's correction.

vessels in $Cnr1^{-/-}Cnr2^{-/-}$ PDZ is associated with misregulated angiogenic factors: *Angpt1*, which stabilizes vasculature, is abnormally expressed in $Cnr1^{-/-}Cnr2^{-/-}$ implantation sites. *Angpt2*, which destabilizes preexisting vasculature for *Angpt1* sprouting (*Maisonpierre et al., 1997*), is downregulated in $Cnr1^{-/-}Cnr2^{-/-}$ implantation sites. Our results also show that *Hif1a* has a similar expression pattern to *Angpt1* in WT PDZ on day 6 of pregnancy, while the *Hif2a* expression pattern overlaps with that of *Angpt2*. This suggests that the deletion of both *Cnr1* and *Cnr2* impacts PDZ formation and establishment of decidualization through regulation of angiogenic and transcriptional factors. The levels of *Cnr1*, which are usually low in stromal cells of non-pregnant females, increase during the decidual process in both humans and mice. Using an in vitro decidualization model, we further showed that decidualization is compromised using $Cnr1^{-/-}$ stromal cells. Consequently, defective PDZ formation results in more resorption later in pregnancy in $Cnr1^{-/-}Cnr2^{-/-}$ females.

Most previous work examining the roles of endocannabinoids in decidualization applied cannabinoids to stromal cells. Although some pharmacological studies using CB1 antagonists suggest CB1 mediates the effects of endocannabinoids on stromal cells, there is no genetic evidence to support this conclusion. Using stromal cells isolated from pregnant rats, a previous report showed that endocannabinoid anandamide (AEA) interferes with stromal differentiation (*Fonseca et al., 2015*). *In vivo* application of AEA in pseudopregnant rats impaired decidualization with reduced levels of Cox1 and VEGF (*Fonseca et al., 2015*). There was evidence for a similar observation in humans. In this respect, AEA impairs cell proliferation and differentiation in the human endometrial stromal cell line and primary cultures of decidual fibroblasts from term pregnancy (*Almada et al., 2016*). There is further evidence that oxidative metabolites of AEA by Cox2 may compromise the normal decidual process (*Almada et al., 2017*; *Almada et al., 2015*).

In contrast to the ectopic application of endocannabinoids, our study used mouse models with suppressed cannabinoid signaling. The studies cited above seem to contradict our current findings that cannabinoid signaling is detrimental to decidualization in rats and humans. However, we have shown previously that either amplified or silenced cannabinoid signaling can interfere with early pregnancy (*Sun and Dey, 2012*). These results suggest that optimal endocannabinoid signaling contributes to pregnancy success, reinforcing our conclusion that normal decidualization requires appropriate cannabinoid signaling.

The function of ANGPT2 in angiogenesis largely depends on the presence of VEGF (*Holash et al., 1999a*; *Holash et al., 1999b*) in that VEGF, ANGPT1 and ANGPT2 function cooperatively to promote angiogenesis (*Asahara et al., 1998*; *Visconti et al., 2002*). ANGPT2 destabilizes the preexisting vasculature, which consequently responds to angiogenic stimuli. The activated vasculature is extended and stabilized by ANGPT1 (*Holash et al., 1999b*). In the absence of VEGF, ANGPT2 destabilizes vasculature, which undergoes vascular regression. In implantation sites, VEGF and ANGPT1 have similar expression patterns on day 6 of pregnancy (*Halder et al., 2000*), but neither is present in the PDZ; only ANGPT2 is expressed in this zone. The consequent avascular zone in the PDZ is considered conducive to the implanting embryo. In the current study, *Angpt1* expression is retained in $Cnr1^{-/-}Cnr2^{-/-}$ PDZ with reduced expression of *Angpt2*, so that vasculature and macrophages are still observed in $Cnr1^{-/-}Cnr2^{-/-}$ PDZs.

Hypoxia-inducible factors (HIFs) are key modulators of the transcriptional responses to hypoxic stress. HIFs are heterodimers of an α-subunit and a stable β-subunit (ARNT). Of the three isoforms, HIF1A and HIF2A are the most studied and structurally similar. HIF1A is considered to have a broader expression pattern in most cells, whereas HIF2A is selectively induced in certain cell types, including endothelial cells and renal interstitial cells (*Keith et al., 2012*). In our study, *Hif1a* is expressed mostly in stromal cells except for the PDZ cells of day 6 implantation sites, while *Hif2a* is induced in PDZ. The transcriptional regulation of HIFs on angiopoietins is specific to tissue and cell-type. For example, in primary human endothelial cells, HIF1A increases mRNA levels of *ANGPT2* and *ANGPT4* but not *ANGPT1* (*Yamakawa et al., 2003*). In the current study, the overlapping expression of *Hif1a* and *Angpt1*, in tandem with the similar expression pattern of *Hif2a* and *Angpt2*, suggests that HIF1A increases *Angpt1* expression in stromal cells and HIF2A induces *Angpt2* transcription in decidualized cells in the PDZ. Taken together, our study illuminates a major responsibility for endocannabinoid signaling in modulating vascular remodeling and integrity. This finding was exemplified in increased vascular leakage at $Cnr1^{-/-}Cnr2^{-/-}$ implantation sites (*Li et al., 2019*). In conclusion, this investigation reveals that CB1 actions in decidual cells, combined with angiogenic

activities driven by CB2, facilitate the formation of the avascular PDZ, which is critical for pregnancy success. Whether other signaling pathways join in this orchestration remains to be seen.

Decidualization is a complex process. Although there are unique differences, some common features of this process are evident in mice and humans. As such, decidualization in mice begins with implantation, whereas decidualization in humans occurs in each menstrual cycle, but becomes more intense like in mice, after implantation (*Cha et al., 2012*). Our results suggest that the induction of CB1 is a common feature shared by mouse and human decidual cells, and blockage of CB1 compromises *in vitro* decidual responses in mice and humans. Whether the PDZ is formed in human implantation or other subhuman primates is not known. However, the epithelial plaque formed during early pregnancy in macaque may have a similar function (*Enders, 2007*).

# Materials and methods

## Key resources table

| Reagent type (species) or resource | Designation | Source or reference | Identifiers | Additional information |
|---|---|---|---|---|
| Antibody | Anti-E-cadherin (Rabbit monoclonal) | Cell Signaling Technology | Cat# 3195S, RRID:AB_2291471 | Whole mount IF(1:100) |
| Antibody | Alexa Fluor 594 AffiniPure Donkey Anti-Rabbit IgG (H+L) | Jackson Immuno Research | Cat# 711-585-152, RRID:AB_2340621 | Whole mount IF(1:300) |
| Antibody | Anti-Cytokeratin 8, TROMA-1 (Rat monoclonal) | Developmental Studies Hybridoma Bank | Cat# TROMA-I, RRID:AB_531826 | IF (1:100) |
| Antibody | Anti-CD45, (Rat monoclonal) | Biolegend | Cat# 103101, RRID:AB_312966 | IF (1:200) |
| Antibody | Anti-HIF1A (Rabbit monoclonal) | Cell Signaling Technology | Cat# 36169T, RRID:AB_2799095 | IF (1:200) |
| Antibody | Anti-HIF2A (Rabbit polyclonal) | Novus | Cat# NB100-122SS, RRID:AB_10002593 | IF (1:500) |
| Antibody | Anti-F4/80 (Rat monoclonal) | Biolegend | Cat# MCA497R, RRID:AB_323279 | IF (1:200) |
| Antibody | Anti-Scrib (Rabbit polyclonal) | Santa Cruz | Cat# sc-28737, RRID:AB_2184807 | IF (1:500) |
| Antibody | Alexa Fluor 594 AffiniPure Donkey Anti-Rat IgG (H+L) | Jackson Immuno Research | Cat#712-585-150, RRID:AB_2340688 | IF (1:400) |
| Antibody | Alexa Fluor 488 AffiniPure Donkey Anti-Rabbit IgG (H+L) | Jackson Immuno Research | Cat# 711-545-152, RRID:AB_2313584 | IF (1:400) |
| Antibody | Cy2 AffiniPure Donkey Anti-Rat IgG (H+L) | Jackson Immuno Research | Cat# 712-225-153, RRID:AB_2340674 | IF (1:400) |
| Biological sample (*M. musculus*) | Primary WT, *Cnr1*−/− and *Cnr2*−/− mouse uterine stromal cells | The current laboratory | | Primary mouse uterine stromal cells freshly isolated from mouse uteri |

*Continued on next page*

*Continued*

| Reagent type (species) or resource | Designation | Source or reference | Identifiers | Additional information |
|---|---|---|---|---|
| Biological sample (*Homo-sapiens*) | Human uterine fibroblasts | Cincinnati Children's Hospital | | A gift from Handwerger laboratory. Aliquots maintained as frozen cells |
| Sequence-based reagent | *Mouse Prl8a2_F* | This paper | PCR primers | ACGTGATGAG GAGGTTCT |
| Sequence-based reagent | *Mouse Prl8a2_R* | This paper | PCR primers | AATCTTGCC CAGTTATGC |
| Sequence-based reagent | *Mouse Cnr1_F* | This paper | PCR primers | CATTGGGACT ATCTTTGCGG |
| Sequence-based reagent | *Mouse Cnr1_R* | This paper | PCR primers | GGTTCTGGAG AACCTGCTGG |
| Sequence-based reagent | *Mouse Rpl7_F* | This paper | PCR primers | GCAGATGTACC GCACTGAGATTC |
| Sequence-based reagent | *Mouse Rpl7_R* | This paper | PCR primers | ACCTTTGGGCT TACTCCATTGATA |
| Sequence-based reagent | *Human CNR1_F* | This paper | PCR primers | gctgcctaaatccactctgc |
| Sequence-based reagent | *Human CNR1_R* | This paper | PCR primers | tggacatga aatggcagaaa |
| Sequence-based reagent | *Human CNR2_F* | This paper | PCR primers | GATTGGCA GCGTGACTATGA |
| Sequence-based reagent | *Human CNR2_R* | This paper | PCR primers | GATTCCGGAA AAGAGGAAGG |
| Sequence-based reagent | *Human IGFBP1_F* | This paper | PCR primers | ccaaactgcaacaagaatg |
| Sequence-based reagent | *Human IGFBP1_R* | This paper | PCR primers | gtagacgcaccagcagag |
| Sequence-based reagent | *Human PRL_F* | This paper | PCR primers | aagctgtagagat tgaggagcaaa |
| Sequence-based reagent | *Human PRL_R* | This paper | PCR primers | tcaggatgaac ctggctgacta |
| Sequence-based reagent | *Human VEGFA_F* | This paper | PCR primers | aaggagga gggcagaatcat |
| Sequence-based reagent | *Human VEGFA_R* | This paper | PCR primers | cacacaggat ggcttgaaga |
| Sequence-based reagent | *Human ANGPT1_F* | This paper | PCR primers | ggacagcagg aaaacagagc |

*Continued on next page*

*Continued*

| Reagent type (species) or resource | Designation | Source or reference | Identifiers | Additional information |
|---|---|---|---|---|
| Sequence-based reagent | *Human ANGPT1_R* | This paper | PCR primers | cacaagcatca aaccaccat |
| Sequence-based reagent | *Human ANGPT2_F* | This paper | PCR primers | ataagcagc atcagccaacc |
| Sequence-based reagent | *Human ANGPT2_R* | This paper | PCR primers | aagttggaag gaccacatgc |
| Sequence-based reagent | *Human HIF1A_F* | This paper | PCR primers | tcatccaagaa gccctaacg |
| Sequence-based reagent | *Human HIF1A_R* | This paper | PCR primers | cgctttctctgagcattctg |
| Sequence-based reagent | *Human HIF2A_F* | This paper | PCR primers | gaacagcaagagcaggttcc |
| Sequence-based reagent | *Human HIF2A_R* | This paper | PCR primers | ggcagcaggtaggactcaaa |
| Sequence-based reagent | *Human GAPDH_F* | This paper | PCR primers | gaaggtgaaggtcggagt |
| Sequence-based reagent | *Human GAPDH_R* | This paper | PCR primers | gatggcaacaatatccactt |
| Chemical compound, drug | *SR141716* | Cayman Chemical | Cat# 9000484 | 2 µM in culture |
| Other | *RPMI 1640 Medium* | Corning | Cat# 15040-cv | |
| Other | Fetal bovine serum | Fisher | Cat# FB12999102 | |

## Animals and treatments

*Cnr1*[-/-] and *Cnr2*[-/-] mice were generated as described (*Zimmer et al., 1999*; *Járai et al., 1999*) and *Cnr1*[-/-]*Cnr2*[-/-] females were generated by crossing *Cnr1*[-/-] and *Cnr2*[-/-] mice. All genetically modified mice and WT controls were maintained on a C57BL6 mixed background and housed in the animal care facility at the Cincinnati Children's Hospital Medical Center according to NIH and institutional guidelines for laboratory animals. All protocols of the present study were approved by the Cincinnati Children's Hospital Research Foundation Institutional Animal Care and Use Committee. All mice were housed in wall-mount negative airflow polycarbonate cages with corn cob bedding. They were provided ad libitum with double distilled autoclaved water and rodent diet (LabDiet 5010).

Female mice were mated with WT fertile males to induce pregnancy (vaginal plug = day 1 of pregnancy). Implantation sites were visualized by an intravenous injection of 0.1 mL of 1% Chicago blue dye solution in saline 4 min before killing, and the number of implantation sites, demarcated by distinct blue bands, was recorded. Mice were euthanized by cervical dislocation right before tissue collection under deep anesthesia. Mice were killed on different days of pregnancy in accordance with the experimental design,that is, day 4, day 6, and day 8 of pregnancy.

## Fluorescence in situ hybridization

In situ hybridization was performed as previously described (*Yuan et al., 2019*). In brief, implantation sites from three individual animals in each experimental group were collected. Frozen sections (12

μm) from three implantation sites from different females in each group were mounted onto poly-L-lysine-coated slides and fixed in 4% paraformaldehyde in PBS. Following acetylation and permeabilization, slides were hybridized with the DIG-labeled *Bmp2*, *Ptgs2*, *Hoxa10*, *Hif1a*, *Hif2a*, *Angpt1*, *Angpt2,* and *Vegfa* probes at 55˚C overnight. After hybridization, slides were then washed, quenched in $H_2O_2$ (3%), and blocked in blocking buffer (1%). Anti-Dig-peroxidase was applied onto hybridized slides and color was developed by Tyramide signal amplification (TSA) Fluorescein according to the manufacturer's instructions (PerkinElmer). Images presented are representative of three independent experiments.

### Immunofluorescence

Implantation sites from three individual animals in each experimental group were snap-frozen. Frozen sections (12 μm) of three implantation sites from different females in each group were mounted onto poly-L-lysine-coated slides and fixed in 4% paraformaldehyde in PBS. Sections were blocked in 5% BSA in PBS and incubated with primary antibody at 4˚C overnight, followed by incubation in secondary antibodies at room temperature for 1 hr in PBS. Nuclear staining was performed using Hoechst 33342 (H1399, Molecular Probes, 2 μg/mL). Immunofluorescence was visualized under a confocal microscope (Nikon Eclipse TE2000). Images presented are representative of three independent experiments. Antibodies for Cytokeratin 8 (Iowa hybridoma bank, 1:100 dilution), CD45 (Biolegend, 1:200 dilution), HIF1A (Cell signaling, 1:200 dilution), HIF2A (Novus, 1:500 dilution), F4/80 (Biorad, 1:200 dilution), and Scrib (Santa Cruz, 1:500 dilution) were used for immunofluorescence staining. All fluorophore-conjugated secondary antibodies (used in 1:400 dilution) were from Jackson Immunoresearch.

### Whole-mount immunostaining for 3D imaging

One side of uterine horns from three WT and *Cnr1*[-/-]*Cnr2*[-/-] females on day 6 of pregnancy were fixed in Dent's Fixative (Methanol: DMSO, 4:1) overnight at −20˚C. After fixation, tissues were then washed in 100% Methanol three times for 1 hr each and bleached with 3% $H_2O_2$ in methanol overnight at 4˚C to eliminate pigmentation. The samples were washed in PBS-T containing 0.1% Tween20 three times for 1 hr each at room temperature and then blocked in 5% BSA in PBS-T overnight at 4˚C. The samples were then incubated with an anti-E-cadherin antibody (1:100, 3195 s, Cell Signaling Technology) on a rotor for 7 d at 4˚C. After incubation, the samples were then washed six times in PBS-T for 1 hr each at room temperature and then incubated with Alexa Fluor 594 AffiniPure Donkey Anti-Rabbit IgG (H + L) (1:300, Jackson ImmunoResearch) on a rotor for 4 d in a light-proof tube at 4˚C. The samples were stored in the dark until tissue clearing.

### Tissue clearing for 3D imaging

The stained samples were held straight with forceps in 100% methanol for 1 min to align the mesometrial–antimesometrial (M–AM) axis and then dehydrated in 100% methanol for 30 min. Dehydration was followed by tissue clearing by BABB (Benzyl alcohol: Benzyl benzoate (1:2); each reagent from Sigma-Aldrich) for 1 hr at room temperature. The samples were then stored in the dark until 3D imaging acquisition (*Yuan et al., 2018*).

### 3D imaging and processing

3D pictures of the implantation site in each processed uterine horn were acquired by a Nikon multiphoton upright confocal microscope (Nikon A1R) with a 10× objective. To obtain the 3D structure of the tissue, the surface tool Imaris (version 9.2.0, Bitplane) was used (*Yuan et al., 2018*). One representative picture of mouse implantation sites in each genotype is presented.

### In vitro decidualization of mouse stromal cells

Stromal cells were collected by enzymatic digestion of mouse uteri on day 4 of pregnancy as described previously (*Sun et al., 2016*). The cells were cultured in phenol-red free DMEM/F12 medium supplemented with charcoal-stripped 1% FBS (*w/v*) overnight before the initiation of decidualization by treatment of estradiol (10 nM) and medroxyprogesterone acetate (1 μM) for 6 d.

## In vitro decidualization of HuF cells

The HuF cells were generated by Stuart Handwerger's laboratory at the Cincinnati Children's Hospital Medical Center under appropriate approval by the Institutional Review Board (*Grinius et al., 2006*). In brief, maternal decidual tissues were dissected from the chorionic layer of human term placentae and enzymatically dispersed (*Richards et al., 1995*). The HuF cells were isolated by differential plating to >95% purity as previously described (*Sherafat-Kazemzadeh et al., 2011*). The purity of isolated cells was confirmed by the expression of Vimentin, a mesenchymal cell marker, and the absence of markers of bone marrow-derived cells and epithelial cells (*Richards et al., 1995*). Handwerger's laboratory generously shared the HuF cells with us for our research on human decidualization. These cells maintained in our laboratory are tested negative for mycoplasma contamination. After reaching 90% confluence, the cells were decidualized by the addition of medroxyprogesterone acetate (1 $\mu$M), estradiol (E$_2$; 10 nM), and prostaglandin E$_2$ (PGE$_2$; 1 $\mu$M) in an RPMI medium that contained 2% fetal bovine serum and antibiotics. The medium in each well was changed at 3 d intervals unless indicated otherwise.

## Quantitative RT-PCR

RNA was collected from WT and *Cnr2$^{-/-}$* uterine samples; WT, *Cnr1$^{-/-}$*, and *Cnr2$^{-/-}$* primary stromal cells; and human uterine fibroblast cells. RNA was analyzed as described previously (*Sun et al., 2014*; *Das et al., 1995*). In brief, total RNA was extracted with Trizol (Invitrogen, USA) according to the manufacturer's protocol. After DNase treatment (Ambion, USA), 1 $\mu$g of total RNA was reverse transcribed with Superscript II (Invitrogen). Real time PCR was performed using primers 5'-ACGTGA TGAGGAGGTTCT-3' (sense) and 5'- AATCTTGCCCAGTTATGC-3' (anti-sense) for mouse *Prl8a2*; 5'-CATTGGGACTATCTTTGCGG-3' (sense) and 5'-GGTTCTGGAGAACCTGCTGG-3' (anti-sense) for mouse *Cnr1*; 5'- GCAGATGTACCGCACTGAGATTC-3' (sense) and 5'-ACCTTTGGGCTTACTCCA TTGATA-3' (anti-sense) for mouse *Rpl7*; 5'- gctgcctaaatccactctgc-3' (sense) and 5'- tggacatgaaatgg cagaaa-3' (anti-sense) for human *CNR1*; 5'- GATTGGCAGCGTGACTATGA-3' (sense) and 5'-GA TTCCGGAAAAGAGGAAGG-3' (anti-sense) for human *CNR2*; 5'-ccaaactgcaacaagaatg-3' (sense) and 5'-gtagacgcaccagcagag-3' (anti-sense) for human *IGFBP1*; 5'- aagctgtagagattgaggagcaaa-3' (sense) and 5'-tcaggatgaacctggctgacta-3' (anti-sense) for human *PRL*; 5'- aaggaggagggcagaatcat-3' (sense) and 5'- cacacaggatggcttgaaga-3' (anti-sense) for human *VEGFA*. 5'- ggacagcaggaaaacagagc-3' (sense) and 5'- cacaagcatcaaaccaccat-3' (anti-sense) for human *ANGPT1*. 5'- ataagcagcatcagccaacc-3' (sense) and 5'- aagttggaaggaccacatgc-3' (anti-sense) for human *ANGPT2*. 5'- tcatccaagaagccc-taacg-3' (sense) and 5'- cgctttctctgagcattctg-3' (anti-sense) for human *HIF1A*. 5'- gaacagcaagag-caggttcc-3' (sense) and 5'- ggcagcaggtaggactcaaa-3' (anti-sense) for human *HIF2A*. 5'-gaaggtgaaggtcggagt-3' (sense) and 5'- gatggcaacaatatccactt-3' (anti-sense) for human *GAPDH*. To examine in vivo Cnr1 levels, RNA was collected from three WT day 4 uterine samples and three different implantation sites from WT and *Cnr2$^{-/-}$* females,respectively, on day 8 of pregnancy. RNAs were collected from mouse primary cell culture in triplicate. In experiments with Huf cells, three samples in each time point were collected to examine *CNR1*, *PRL,* and *IGFBP1* levels. To study the effects of SR1 on decidualization in Huf cells, five samples were collected from each treatment group. Sample sizes are indicated in the figures.

## Statistical analysis

The number of macrophages are counted using three sections from three different animals. F4/80 positive cells within endometrial domains are quantified using three sections from three WT and three *Cnr1$^{-/-}$Cnr2$^{-/-}$* females on day 6 of pregnancy. Data were analyzed using *t*-tests as indicated in figure legends. Data are shown as mean ± SEM. All *P* values are depicted in the figure legends.

## Acknowledgements

We thank Katie Gerhardt for her efficient editing of the manuscript. This work was supported in parts by NIH grants (HD103475 and HD068524 to SKD).

## Additional information

### Funding

| Funder | Grant reference number | Author |
| --- | --- | --- |
| National Institute on Drug Abuse | DA006668 | Sudhansu K Dey |
| Eunice Kennedy Shriver National Institute of Child Health and Human Development | HD068524 | Sudhansu K Dey |
| Eunice Kennedy Shriver National Institute of Child Health and Human Development | HD103475 | Sudhansu K Dey |

The funders had no role in study design, data collection and interpretation, or the decision to submit the work for publication.

### Author contributions

Yingju Li, Amanda Dewar, Yeon Sun Kim, Data curation, Formal analysis, Investigation; Sudhansu K Dey, Conceptualization, Formal analysis, Supervision, Funding acquisition, Investigation, Methodology, Writing - review and editing; Xiaofei Sun, Conceptualization, Data curation, Formal analysis, Supervision, Investigation, Methodology, Writing - original draft, Project administration, Writing - review and editing

### Author ORCIDs

Sudhansu K Dey https://orcid.org/0000-0001-9159-186X
Xiaofei Sun https://orcid.org/0000-0001-9601-5423

### Decision letter and Author response

Decision letter https://doi.org/10.7554/eLife.61762.sa1
Author response https://doi.org/10.7554/eLife.61762.sa2

## Additional files

### Supplementary files

- Transparent reporting form

### Data availability

All data generated or analysed during this study are included in the manuscript and supporting files.

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
