## [Decision Letter]

**Acceptance summary:**

With the rapid expansion of cannabis use, understanding the effects of this drug on pregnancy is becoming increasingly vital. In this regard, the results of work such as yours could be used by regulatory agencies, for example, in deciding about information to include on warning labels. Thus, the data in your paper has important scientific and practical value.

**Decision letter after peer review:**

[Editors’ note: the authors submitted for reconsideration following the decision after peer review. What follows is the decision letter after the first round of review.]

Thank you for submitting your work entitled "Pregnancy success in mice requires appropriate cannabinoid receptor signaling for primary decidua formation" for consideration by *eLife*. Your article has been reviewed by three peer reviewers, one of whom is a member of our Board of Reviewing Editors, and the evaluation has been overseen by a Senior Editor. The reviewers have opted to remain anonymous.

Our decision has been reached after consultation among the reviewers. Based on these discussions and the individual reviews below, we regret to inform you that your work will not be considered further for publication in *eLife*.

All the reviewers found the work submitted for publication to be important and interesting. However, the suggested experiments would take significant time to perform.

Reviewer #1:

Previously the investigators showed that *Cnr1^-/-^/Cnr2^-/-^* mice exhibit defective decidualization and are subfertile. Here they go on to explore the mechanism behind aberrant decidualization. They use a series of marker genes to show that the defect emerges during the postimplantation period in vivo and is attributable to malformation of the primary decidual zone. They also show that *Cnr1* is up regulated during and required for decidualization of mouse uterine stromal cells in vitro. Similarly, decidualization of human uterine stromal cells is accompanied by very substantial increases in CNR1 expression. A major strength of this work is the importance of understanding the role of these receptors in reproduction given the widespread use of marijuana. Weaknesses include the fact that many of the results are based on correlatives rather than direct functional relationships.

1) The number of animals that were employed for each experiment should be clearly stated.

2) Whenever possible more quantitative analyses would be desirable, for example, immunoblots to confirm altered expression of the proteins encoded by the marker genes. Added proof for abnormal vascularization of the primary decidual zone in the form of counting the vessels would be desirable. Similarly, quantifying macrophage retention would also be helpful. Also, the degree of contraction or expansion of marker expression could be measured.

3) The functional relationships that are drawn among Vegf, the Hifs and the Ang1/2 in situ hybridization patterns should be softened as Hif activity is regulated at the protein level.

4) The in vitro analyses are somewhat superficial. In addition to assaying the expression of molecules that correlate with decidualization, pictures of the cells that document the expected morphological changes would also be desirable.

5) The authors show a very impressive rise in CNR1 expression with decidualization of human uterine fibroblasts. However, they fail to show the importance of this receptor in the process by demonstrating the effects of its elimination.

6) The starting material for both the mouse and human in vitro experiments appeared to be decidua. Were the cells de-differentiated and how many different donors were employed?

Reviewer #2:

The manuscript by Li and coworkers investigates the role of cannabinoid receptor signaling in pregnancy. The tow cannabinoid receptors CB1 and CB2 are expressed in uterine stroma cells and endothelial cells, respectively. They show that the double knockout of CB1 and CB2 result in impaired primary decidual zone development, increase invasion of immune cells and increased vascularization. They also show that CB1 is increased in human uterine fibroblasts that decidualize in vitro. They imply that the two receptor must coordinate the uterine fibroblast differentiation and endothelial cells act together for normal embryo implantation. This is an exciting finding and will generate general interest. However there are two issues that would strengthen this manuscript.

The manuscript should clearly define the phenotype of the single knockouts of CB1 and CB2 to show which parts of the double phenotype are present in the single and double and which phenotypes are novel with respect to the double knockout. This is with respect the expression of the marker genes observed. This would show that they separately regulate the vascularization and stroma cell decidualization and together these process are required or is there synergy between these two compartments. This should be discussed.

The authors imply a role of CB1 in human endometrial stroma fibroblast decidualization. They should employ knockdown of CB1 in these cells and show the impact on decidualization. This would increase the relevance of this manuscript.

Reviewer #3:

The manuscript entitled "Pregnancy success in mice requires appropriate cannabinoid receptor signaling for primary decidua formation" by Li Y et al. evaluates the influence of cannabinoid signaling through *Cnr1* and *Cnr2* by forming primary decidual zone (PDZ) along with changes in angiogenic features in the decidual bed during pregnancy. The deficiency of these genes in mice, results in compromised primary decidualization, since it presents abnormal blood vessels formation and presence of macrophages into the implantation chambers.

The hypothesis that cannabinoid/endocannabinoid signalling, through *Cnr1* and *Cnr2*, compromises decidualization and embryo implantation success is straightforward and the double mutant model *Cnr1^-/-^ Cnr2^-/-^* used is appropriate. However, the experimental design used in each experiment is confuse and not consistent. The results explanations and the figures are a quite difficult to understand. There is a lack of detail about the experimental planing and the sample size used. The relevance and interpretation for the inclusion of in vitro decidualization of human uterine fibroblast experiment is not explained properly.

This reviewer has the following major comments and suggestions:

– For consistency, this work uses three different mice models *Cnr1^-/-^, Cnr2^-/-^* and double mutant *Cnr1^-/-^Cnr2^-/-^.* However, it is very confusing to follow which model is used in each experimental part and the reason to do it. The authors should detail the mouse model used for each experiment and to justify their use.

– The experimental design is missing and should be included along the manuscript in the Results, Materials and methods, and figure legends. To understand this work the readers need to know mouse model or human samples used, sample size and replicates in each experiment. Those experiments performed using mice models should be included: days of female sacrifice (day 4, 5, 6 and 8), sample size for each sacrifice day, tissue collected, number of implantation sites analyzed, number of sections evaluated in each experiment of immunofluorescence and in situ hybridization. These details should be included along the manuscript to understand the results presented.

– The authors indicate along the manuscript that their finding about primary decidual zone formation is disrupted in *Cnr1^-/-^Cnr2^-/-^* females compromising their pregnancy outcomes. However, no data or result about pregnancy outcome is included. Introduction: "In this study, we show that endothelial cells deleted of *Cnr2* have important roles in embryo implantation"; "Cnr1's critical role in early pregnancy", "The results show that double mutant mice have compromised PDZ formation, resulting in suboptimal pregnancy outcome". However, no data or result about pregnancy outcome is included.

– Decidualization is the core process of the study. The authors should explain the biology of this process and its role in pregnancy in the Introduction section. In addition, the Introduction did not mention that the in vitro approach with human uterine fibroblast; however it is mentioned in the Abstract. Please include these experiments in the Introduction and its connection with the rest of the results.

– The experimental techniques are based on the use of fluorescence in situ hybridization and immunofluorescence. These experiments let us to observe the expression pattern and signal intensity between groups (wild type and *Cnr1^-/-^Cnr2^-/-^* mice). However, all the findings showed are descriptive. The author should include a quantitative method to quantify these parameters and to apply a statistical analysis to corroborate if the differences observed are significant. It is crucial to strengthen the results.

– Materials and methods shows a description about RT-PCR but no experiment was included along the manuscript. It seems that these experiments were performed but are missing in the figures. Please, include this data that are very useful to corroborate the fluorescence in situ hybridization results, using a quantitative method as RT-PCR.

– The IF and in situ hybridization experiments should be performed using the same markers for the different days selected. For example, in the Figure 3A the staining CD45 and F4/F80 was performed in day 4 of pregnancy and Figure 3B F4/F80 and PECAM in day 6 of pregnancy. Please, show the same markers for the different days analysed.

[Editors’ note: further revisions were suggested prior to acceptance, as described below.]

Thank you for resubmitting your work entitled "Pregnancy success in mice requires appropriate cannabinoid receptor signaling for primary decidua formation" for further consideration by *eLife*. Your revised article has been evaluated by Didier Stainier (Senior Editor) and a Reviewing Editor.

The manuscript has been improved but there are some remaining issues that need to be addressed before acceptance, as outlined below in the individual reviews:

Reviewer #1:

The authors did a good job of responding to my comments.

Reviewer #2:

The manuscript entitled "Pregnancy success in mice requires appropriate cannabinoid receptor signaling for primary decidua formation" by Li Y et al. evaluates the influence of cannabinoid signaling through *Cnr1* and *Cnr2* by forming primary decidual zone (PDZ) along with changes in angiogenic features in the decidual bed during pregnancy. The deficiency of these genes in mice, results in compromised primary decidualization, since it presents abnormal blood vessels formation and presence of macrophages into the implantation chambers.

The authors have followed most of my recommendations and the paper has improved remarkably. Still some points need to be addressed:

– The authors use the sentence "samples sizes are indicated in each figure" in material and methods sections. But the author should include these important data in Materials and methods section.

– The introductory sentence in each figure legend is confused "All images are representative of three independent experiments". How many animals? Human samples? And how many replicates of each experiment, number of tissue sections/cells… analyzed from same animal/patient? There is a lack of detail about experimental design used. Supplementary figure with a scheme would be interesting to understand better this part.

– "In each experiment, three to four animals were used to validate the results". The author should indicate the exact number of mice used from each animal: wild type, *Cnr1^-/-^, Cnr2^-/-^* and *Cnr1^-/-^Cnr2^-/-^* mice.

– The Figure 3 is similar to the last version, just PECAM results was excluded. The authors should include CD45 staining for day 6.

– The authors explain that The *Cnr1^-/-^* null model is now included in the text but the information is missing.

– "Mice were killed on different days of pregnancy in accordance to experimental design. i.e day 4, day 6, and day 8 of pregnancy. In each experiment, three to four animals were used to validate the results." Please define exactly the numbers.

– Subsection “Immunofluorescence.”: tissue sections size?

– Sections were blocked in 5% BSA in PBS and incubated with primary antibody at 4{degree sign}C overnight, followed by incubation in secondary antibody for 1 hour in PBS. There is missing data, Temperature? What dilution media of primary and secondary antibody was used? Please, specify.

In conclusion, the manuscript has improved remarkably, but still more details about experimental design, Materials and methods experiments, numbers of mice and human samples used, number of tissue sections, cells analyzed in each experiment and strategies used to evaluate decidualization status.

Reviewer #3:

This manuscript investigates the role of the cannabinoid receptors *Cnr1* and *Cnr2* in mouse uterine function and human uterine fibroblast decidualization. This manuscript clearly demonstrates that ablation of both receptors impairs pregnancy by altering uterine stroma cell decidualization, vascularization and invasion of macrophages. Using in vitro decidualization of uterine stroma cells from *Cnr1* mull mice and inhibition of cannabinoid signaling in human uterine fibroblasts this manuscript shows that alteration of *Cnr1* signaling impairs mouse and human stroma cell decidualization. This manuscript is a comprehensive evaluation of the role of these receptors in uterine function. The authors have addressed all the concerns of the initial review and teh data presented will advance the field.

---

## [Author Response]

[Editors’ note: the authors resubmitted a revised version of the paper for consideration. What follows is the authors’ response to the first round of review.]

Reviewer #1:[…] A major strength of this work is the importance of understanding the role of these receptors in reproduction given the widespread use of marijuana. Weaknesses include the fact that many of the results are based on correlatives rather than direct functional relationships.1) The number of animals that were employed for each experiment should be clearly stated.

The numbers of animals/samples in immunostaining and *in situ* hybridization experiments are now included in the revised figure legends (Figures 1-5). Sample sizes in quantitative PCR experiments are labeled in the bar diagrams (Figures 6-8).

2) Whenever possible more quantitative analyses would be desirable, for example, immunoblots to confirm altered expression of the proteins encoded by the marker genes. Added proof for abnormal vascularization of the primary decidual zone in the form of counting the vessels would be desirable. Similarly, quantifying macrophage retention would also be helpful. Also, the degree of contraction or expansion of marker expression could be measured.

As suggested, we have performed quantitative analyses of several parameters. Counting blood vessels in uterine sections may not provide more meaningful information, because of changes in the limited domain of the primary decidual zone (PDZ) and due to different dimensions and sizes of uteri under different experimental conditions. Therefore, we have quantified blood vessels in the (PDZ) (Figure 4—figure supplement 1). We have also quantified macrophage numbers in the endometrial domain (Figure 3D), as well as the suppression of Ang1 and Hif1a and induction of Ang2 and Hif2a signals in the PDZ (Figure 5—figure supplement 1). To address the reviewer’s concern regarding decidualization of Huf cells, morphological changes and decidual markers have been quantified (Figure 7 and Figure 7—figure supplement 1). To study the biological significance of CB1 induction of decidualization in Huf cells, we also added quantitative data of changes in angiogenic factors in human uterine fibroblasts before and after decidualization (Figure 8).

3) The functional relationships that are drawn among Vegf, the Hifs and the Ang1/2 in situ hybridization patterns should be softened as Hif activity is regulated at the protein level.

We now provide the expression status of Hif proteins by immunofluorescence staining (Figures 5E and 5F), which corroborates the results of *in situ* hybridization. We have softened the functional relationship statement in the text of the revised manuscript (end of subsection “Retention of blood vessels in Cnr1^-/-^Cnr2^-/-^ PDZ is associated with multiple angiogenic factors”).

4) The in vitro analyses are somewhat superficial. In addition to assaying the expression of molecules that correlate with decidualization, pictures of the cells that document the expected morphological changes would also be desirable.

We have added morphological images of Huf cells before and after decidualization (Figure 7D-F and Figure 7—figure supplement 1). To strengthen the results of in vitro analysis and address the function of CB1 in decidualization, the latter was suppressed by a CB1 receptor specific antagonist (SR141716A, SR1) during decidualization, and the changes in decidual responses, as well as angiogenic factors were examined (Figure 8). More details regarding this question are included in the next question.

5) The authors show a very impressive rise in CNR1 expression with decidualization of human uterine fibroblasts. However, they fail to show the importance of this receptor in the process by demonstrating the effects of its elimination.

This is a legitimate question raised by the reviewer. To show the function of CB1 in decidualization, we used pharmacological inhibition through a specific CB1 receptor antagonist SR1. We found that neutralizing CB1 greatly reduces decidual response with respect to its marker genes and other angiogenic factors (Figure 8).

6) The starting material for both the mouse and human in vitro experiments appeared to be decidua. Were the cells de-differentiated and how many different donors were employed?

For mice, we used day 4 pregnant uteri to isolate purified stromal cells. For human experiments, we used uterine fibroblast cells purified from endometrial tissues obtained from women with typical menstrual cycle; predecidualization stromal cells were undifferentiated, but with decidualization cells are differentiated into decidual cells as indicated by decidual markers. Three donor mice were used in each group (Figure 6). Human uterine fibroblast cell line was generated by Dr. Stuart Handwerger and generously donated to us for future studies (PMID 7506205).

Reviewer #2:The manuscript should clearly define the phenotype of the single knockouts of CB1 and CB2 to show which parts of the double phenotype are present in the single and double and which phenotypes are novel with respect to the double knockout. This is with respect the expression of the marker genes observed. This would show that they separately regulate the vascularization and stroma cell decidualization and together these process are required or is there synergy between these two compartments. This should be discussed.

As suggested by the reviewer, the phenotypic differences between single and double knockout (*Cnr1^-/-^Cnr2^-/-^)* mice are now shown in Figure 5. Decidual defects observed in *Cnr1^-/-^Cnr2^-/-^* mice are observed in *Cnr1^-/-^* mice as well, but not in *Cnr2^-/-^* mice, suggesting that normal decidualization requires CB1.

The authors imply a role of CB1 in human endometrial stroma fibroblast decidualization. They should employ knockdown of CB1 in these cells and show the impact on decidualization. This would increase the relevance of this manuscript.

Thank you for the constructive suggestion. We have responded to this question using a specific CB1 receptor antagonist SR1 (see our response to reviewer 1’s question 5.)

Reviewer #3:This reviewer has the following major comments and suggestions:– For consistency, this work uses three different mice models Cnr1^-/-^, Cnr2^-/-^ and double mutant Cnr1^-/-^Cnr2^-/-^. However, it is very confusing to follow which model is used in each experimental part and the reason to do it. The authors should detail the mouse model used for each experiment and to justify their use.

The mouse models and the justification for use are now described in the Results section. To mechanistically address the role of CB1 and CB2 in vascular permeability and stromal decidualization, we have added data of angiogenic factors in *Cnr1^-/-^* and *Cnr2^-/-^* mice (Figure 5). Decidual defects observed in *Cnr1^-/-^Cnr2^-/-^* mice are observed in *Cnr1^-/-^* mice as well, but not in *Cnr2^-/-^* mice, which suggests that normal stromal decidualization depends on CB1.

– The experimental design is missing and should be included along the manuscript in the Results, Materials and methods, and figure legends. To understand this work the readers need to know mouse model or human samples used, sample size and replicates in each experiment. Those experiments performed using mice models should be included: days of female sacrifice (day 4, 5, 6 and 8), sample size for each sacrifice day, tissue collected, number of implantation sites analyzed, number of sections evaluated in each experiment of immunofluorescence and in situ hybridization. These details should be included along the manuscript to understand the results presented.

A part of the question is similar to questions 1 and 6 of reviewer 1, please refer to our responses under each of the questions. We have added a full description of the experimental design encompassing the work done in this study.

– The authors indicate along the manuscript that their finding about primary decidual zone formation is disrupted in Cnr1^-/-^Cnr2^-/-^ females compromising their pregnancy outcomes. However, no data or result about pregnancy outcome is included. Introduction: "In this study, we show that endothelial cells deleted of Cnr2 have important roles in embryo implantation"; "Cnr1's critical role in early pregnancy", "The results show that double mutant mice have compromised PDZ formation, resulting in suboptimal pregnancy outcome". However, no data or result about pregnancy outcome is included.

The original data for pregnancy outcomes of each mouse model used in the current study were presented in Figure 1A-1C in the previous report (PMID: 30776303). As suggested by the reviewer, the pregnancy outcomes are now summarized in the manuscript (Introduction section).

– Decidualization is the core process of the study. The authors should explain the biology of this process and its role in pregnancy in the Introduction section. In addition, the Introduction did not mention that the in vitro approach with human uterine fibroblast; however it is mentioned in the Abstract. Please include these experiments in the Introduction and its connection with the rest of the results.

This is a legitimate question and the information is now provided in the Introduction section. The approaches to differentiate human uterine fibroblasts are now included in the Materials and method section.

– The experimental techniques are based on the use of fluorescence in situ hybridization and immunofluorescence. These experiments let us to observe the expression pattern and signal intensity between groups (wild type and Cnr1^-/-^Cnr2^-/-^ mice). However, all the findings showed are descriptive. The author should include a quantitative method to quantify these parameters and to apply a statistical analysis to corroborate if the differences observed are significant. It is crucial to strengthen the results.

We have used semi-quantitative methods to bolster our results and observations with statistical analyses (Figure 5—figure supplement 1) and subsequent interpretation. In addition, we have added more quantitative data using qPCR experiments to examine the role of CB1 during decidualization (Figure 8).

– Materials and methods shows a description about RT-PCR but no experiment was included along the manuscript. It seems that these experiments were performed but are missing in the figures. Please, include this data that are very useful to corroborate the fluorescence in situ hybridization results, using a quantitative method as RT-PCR.

The data in Figures 6 and original Figure 7 were generated using qPCR. As suggested, to confirm the in situ hybridization results, we performed protein immunofluorescence staining (Figure 5E and F) and qPCR of angiogenic factors in HuF cells before and after decidualization (Figure 8).

– The IF and in situ hybridization experiments should be performed using the same markers for the different days selected. For example, in the Figure 3A the staining CD45 and F4/F80 was performed in day 4 of pregnancy and Figure 3B F4/F80 and PECAM in day 6 of pregnancy. Please, show the same markers for the different days analysed.

As suggested, we have included additional data of these markers on different days of pregnancy, except PECAM which is now excluded from the manuscript.

[Editors’ note: what follows is the authors’ response to the second round of review.]

Reviewer #2:The authors have followed most of my recommendations and the paper has improved remarkably. Still some points need to be addressed:– The authors use the sentence "samples sizes are indicated in each figure" in material and methods sections. But the author should include these important data in Materials and methods section.

The number of animals and sample sizes for each experiment are specified in individual method sections, including “Whole-mount immunostaining”, “3D imaging and processing”, “Quantitative RT-PCR”, “Fluorescence *in situ* hybridization” and “Immunofluorescence”.

– The introductory sentence in each figure legend is confused "All images are representative of three independent experiments". How many animals? Human samples? And how many replicates of each experiment, number of tissue sections/cells… analyzed from same animal/patient? There is a lack of detail about experimental design used. Supplementary figure with a scheme would be interesting to understand better this part.

Detailed responses are listed under the first question of review 2.

– "In each experiment, three to four animals were used to validate the results". The author should indicate the exact number of mice used from each animal: wild type, Cnr1^-/-^, Cnr2^-/-^ and Cnr1^-/-^Cnr2^-/-^ mice.

The number of animal and sections used in each experiment are now specified in each method section. Detailed responses are listed under the first question of reviewer 2.

– The Figure 3 is similar to the last version, just PECAM results was excluded. The authors should include CD45 staining for day 6.

We now include data of CD45 of day 6 implantation sites in Figure 3—figure supplement 1.

– The authors explain that The Cnr1^-/-^ null model is now included in the text but the information is missing.

The original concern was about Figure 6C regarding in vitro decidualization using primary stromal cells retrieved from WT, *Cnr1*^-/-^ and *Cnr2*^-/-^ mice. Therefore, in the first sentence in the method section under “Quantitative RT-PCR”, we now state that “RNA was collected from….. WT, *Cnr1^-/-^*and *Cnr2^-/-^*primary stromal cells”.

– "Mice were killed on different days of pregnancy in accordance to experimental design. i.e day 4, day 6, and day 8 of pregnancy. In each experiment, three to four animals were used to validate the results." Please define exactly the numbers.

The number of animals and sections used in each experiment are specified in each method section.

– Subsection “Immunofluorescence.”: tissue sections size?

Frozen sections were cut with 12 μm in thickness. The information is now inserted.

– Sections were blocked in 5% BSA in PBS and incubated with primary antibody at 4{degree sign}C overnight, followed by incubation in secondary antibody for 1 hour in PBS. There is missing data, Temperature? What dilution media of primary and secondary antibody was used? Please, specify.

Incubation of secondary antibodies was conducted at room temperature. The information is now added. Dilution factors for each antibody are specified at the end of method section “Immunofluorescence”.